# Collaborative Effect of In-Plasma Catalysis with Sequential Na₂SO₃ Wet Scrubbing on Co-Elimination of NOx and VOCs from Simulated Sinter Flue Gas

**Juexiu Li** [1,*] , **Rui Zhao** [1], **Maiqi Sun** [2], **Qixu Shi** [1], **Mingzhu Zhao** [1], **Junmei Zhang** [1], **Yue Liu** [1] and **Jinping Jia** [3]

1  School of Energy and Environment, Zhongyuan University of Technology, Zhengzhou 450007, China
2  International Education College, Henan Agricultural University, Zhengzhou 450002, China
3  School of Environmental Science and Engineering, Shanghai Jiao Tong University, Shanghai 200240, China
*  Correspondence: 6858@zut.edu.cn; Tel.: +86-0371-62506813

**Abstract:** Sinter flue gas produced by the iron-ore sinter process in steel plants is characterized by a large gas volume and complex components. Among the major air pollutants, preliminary emissions of volatile organic compounds (VOCs) and nitrogen oxides (NOx) exhibit an inevitable contribution to secondary aerosol and ozone formation. Herein, oxidation–absorption collaborative technology for in-plasma catalysis with sequential Na₂SO₃ wet scrubbing, aiming at co-elimination of NOx and VOCs from sinter flue gas, is proposed. Experimental parameters, including plasma discharge status, NO initial concentration, gas feed flux, Na₂SO₃ concentration, pH value, and absorption ions, were systematically investigated. The VOC and NOx removal performance of the integrated system was further investigated by taking simulated sinter flue gas as model pollutants. The results showed that the collaborative system has satisfactory performance for TVOC and NO removal rates for the effective oxidation of in-plasma catalysis and Na₂SO₃ absorption. The integration of plasma catalysis with Na₂SO₃ scrubbing could be an alternative technology for the co-elimination of sinter flue gas multi-compounds.

**Keywords:** sinter flue gas; volatile organic compounds; in-plasma catalytic; nitrogen oxides; Na₂SO₃; integration





## 1. Introduction

Air-pollutant emissions from anthropogenic sources have contributed to severe hazards for atmospheric pollution and human health on a global scale and especially in China in recent decades [1–3]. Despite the positive collaborative governance effort towards haze pollution reduction [4,5], China still faces the co-existing threats of PM$_{2.5}$ and O$_3$ pollution during the current 14th five-year plan period. Among the main anthropogenic sources, the iron and steel industry is a major primary atmospheric pollutant emission source, owing to its intensive energy and material consumption. As well as being the largest global iron and steel producer, China's annual crude steel production reached 1.03 billion tons in 2022 and has accounted for more than 50% of the global total output for decades. The current air-pollutant emission amount from the iron and steel industry have exceeded that of thermal power plants [6]. It has also been identified that haze outbreaks and air-quality improvement concurrently occur with steel-making activities [7–9].

The iron-ore sinter process, with high dependence on fossil fuel, negative suction combustion, and massive air consumption, accounts for the largest proportion of steel-making emissions [10–12]. The components of sinter flue gas are also complicated, including particulate matter (PM) [13,14], SO$_2$ [15,16], NOx [17], VOCs [18], heavy metals [19], dioxins [20], and trace elements [21]. The ultra-low emission policy of the Ministry of Ecology and Environment of China regulates sinter flue gas PM, SO$_2$, and NOx emissions as having to be less than 10, 35, and 50 mg·m$^{-3}$. Our previous study

revealed and investigated the simultaneous emissions of VOCs and NO during the sintering process [22]. Moreover, NOx can contribute to photochemical smog and $O_3$ formation when co-existing with VOCs under sunlight irradiation via a series of free-radical reactions [23,24]. The massive primary emissions of NOx and VOCs from sinter flue gas on secondary aerosol formation are much less studied, and their total impact may be underestimated. Therefore, NOx and VOCs exhibit huge deduction potential among the major air pollutants of sinter flue gas. It is urgently necessary to develop advanced elimination techniques aiming at the simultaneous abatement of NOx and VOCs from sinter flue gas emissions before the primary emission from sinter exhausts and to develop multipollutant control technology.

Various measures have been widely applied to decrease sinter flue gas primary emissions, such as desulfurization, denitrification facilities, and electrostatic precipitators. However, whether the VOC emissions can be jointly controlled by the measures taken to control PM, $SO_2$, and NOx emissions in sinter flue gas is not understood. Currently, primary flue gas denitrification (deNOx) technology includes selective catalytic reduction (SCR) [25], wet and dry scrubbing [26], adsorption [27], and biological treatments [28]. As for VOC removal methods, adsorption [29], biotechnology [30], catalytic oxidation (combustion) [31,32], and non-thermal plasma [33] have been developed. Both SCR and catalytic oxidation technologies play essential roles in flue gas deNOx and VOC elimination for high efficiency and fewer secondary byproducts. However, for coal-fired flue gas VOC components, the conventional pollutant control process (SCR and WFGD) has only a limited effect on VOC reduction [34]. Moreover, research concerning the co-elimination of VOCs and NOx from sinter flue gas is scarce. The modified $V_2O_5$-$WO_3$/$TiO_2$ (VWT) SCR catalyst is feasible for the simultaneous removal of NO and VOCs within the reaction range of 260–420 °C [35]. Xiao et al. constructed a Cu-VWT bifunctional catalyst for deep oxidation of VOCs and the simultaneous removal of NOx under complex coal-fired flue gas conditions [36]. The removal efficiency of toluene, propylene, dichloromethane, and naphthalene exceeded 99% under 350 °C. Chen et al. developed Ce [37], Ce/Mo [35], and Cu, Fe, and Co [38] modified VWT catalysts that exhibited the collaborative removal of NO and VOCs (benzene and toluene) with different calcination temperatures and transition metal loadings. However, the desirable NOx and VOC removal performance required a specific temperature range (usually >300 °C) that was much higher than the temperature of typical sinter flue gas. Therefore, the sinter flue gas exhaust temperature could not meet the requirements of the SCR and the catalytic combustion temperature, and additional heating energy consumption was needed, which increased initial investment and operating-energy consumption. There is an urgent need to develop deNOx and VOC elimination technology suitable for low-temperature sinter flue gas.

Wet NOx removal technology has a low operating temperature and can simultaneously remove soluble volatile compounds and $SO_2$. More than 90% of the sinter flue gas NOx emissions are NO with poor solubility. Therefore, efficient NOx removal requires the efficient oxidation of NO to $NO_2$, capable of absorption by alkaline and reductive absorption. Non-thermal plasma (NTP) is an advanced oxidation method that can effectively oxidize NO and VOCs at mild operating temperatures and has the advantages of easy start-up and non-selectivity [39]. The combination of NTP and heterogeneous catalysis (the plasma-driven catalysis reaction) can improve catalytic oxidation efficiency. Different catalysts were investigated to enhance VOC removal efficiency and to inhibit byproducts [40–43]. Some other integration techniques have also been investigated for different industrial applications. Zhang et al. [44] developed an integrated system of a spray tower and photocatalysis and applied it to purify the waste gas emitted from a printed circuit board (PCB) manufacturing facility. The integrated technique achieved an average removal efficiency (RE) of 72.39% of 66 VOCs during the nine-month continuous treatment.

As discussed above, to tackle VOC and NOx emissions in the sintering process, it is necessary to design a new strategy to accomplish the simultaneous abatement of VOCs and NOx [45]. In this study, we develop an integration system of in-plasma catalysis (IPC) with sequential $Na_2SO_3$ wet scrubbing for the co-elimination of NOx and VOCs from sinter flue gas. NO can be fast and effectively oxidized to $NO_2$ in the IPC region and further absorbed by $Na_2SO_3$ wet scrubbing. Experimental parameters, including plasma discharge status, NO initial concentration, gas feed flux, $Na_2SO_3$ concentration, pH value, and absorption ions, were systematically investigated. The VOC and NO removal performance of the integrated system was further investigated by taking simulated sinter flue gas as model pollutants.

## 2. Materials and Methods

### 2.1. Chemicals and Reagents

The reagents used in the experiments are all of analytical grade without further purification. In short, sodium sulfite ($Na_2SO_3$) and ascorbic acid ($C_6H_8O_6$) were bought from Shanghai Macklin Biochemical Co., Ltd. (Shanghai, China). Gas feed, including nitrogen monoxide, nitrogen dioxide, and sulfur dioxide with purity $\geq$ 99.999%, was bought from Shanghai Weichuang Co., Ltd. (Shanghai, China). Copper foam with purity $\geq$ 99.8% was bought from Kunshan Jiayisheng Electronics Co., Ltd. (Kunshan, China). Iron-ore sinter raw material, composed of iron-bearing materials, fluxes, and solid fuel, was provided by Taiyuan Iron & Steel Co., Ltd. (TISCO, Taiyuan, China) to generate simulated sinter flue gas.

### 2.2. Experimental Setup and Analytical Method

The experiment setup (Figure 1) was divided into three units with different functions. Specifically, gas feed, controlled via a mass flow meter (pure air containing VOCs, NOx, and $SO_2$) after mixing in Chamber 1, was purged into a plasma catalytic reactor, followed by sequential $Na_2SO_3$ scrubbing. Another gas feed was provided by the air pushing the sinter raw mix into the center of a tubular furnace under different sinter temperatures to generate simulated sinter flue gas containing VOCs, NO, and $SO_2$ components under practical sinter conditions. After thorough mixing in Chamber 2, the simulating flue gas was purged into the IPC unit and tail $Na_2SO_3$ scrubbing unit.

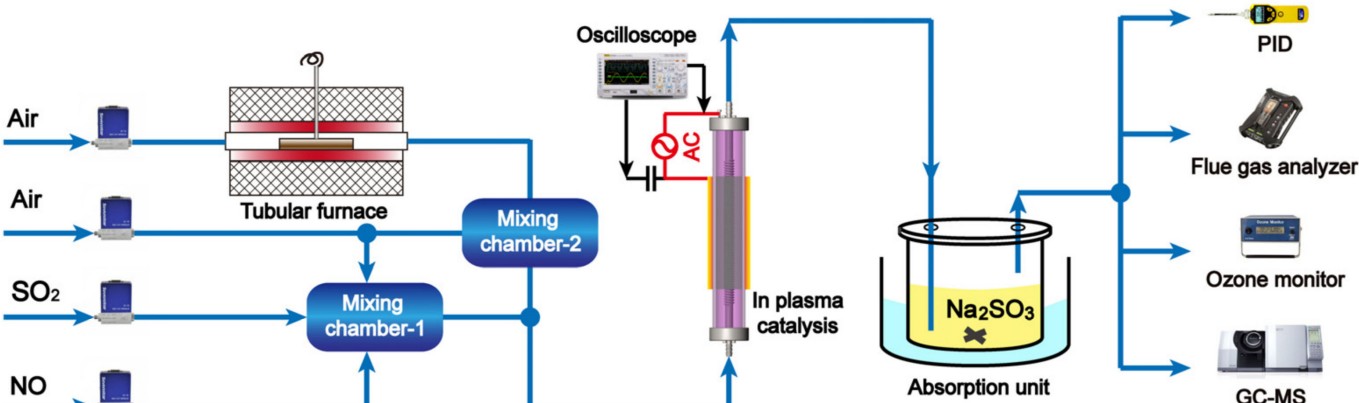

**Figure 1.** Experimental setups for co-elimination of VOCs and NOx from sinter flue gas.

The in-plasma catalytic unit was constructed by a typical coaxial dielectric barrier discharge (DBD) reactor, which consisted of a quartz tube and CuO foam catalyst (Figure S1 in the Supplementary Materials). Specifically, a tubular quartz tube (>99.9% $SiO_2$, dielectric constant: 3.75) with a length of 300 mm, inner diameter of 20 mm, and wall thickness of 2.5 mm was the discharge barrier. A wedged stainless-steel rod with a diameter of 14 mm was end-fixed along the axis of the cylinder and acted as a high-voltage electrode. A stainless-steel mesh with a length of 15 cm wrapped outside the quartz tube acted as

a ground electrode to achieve a discharge volume of 24.0 cm$^3$. The in-plasma discharge gap was filled with monolithic CuO foam (length = 15 mm, thickness = 3 mm), which was fabricated by the calcination of copper foam before being manually rolled into a hollow cylindrical shape, as reported in our previous work [46].

The plasma-generating power supply (AC in Figure 1) was further illustrated in Figure S2 in the Supplementary Materials. Using a high-voltage alternating current (AC) power supply (CTP-2000K, Nanjing Suman Electronics Corp., Nanjing, China) equipped with an amplifier, the employed peak voltage of gas discharge can be adjusted. The maximum voltage of the power supply was 30 kV, and the frequency can vary between 1 kHz to 100 kHz. A high-voltage probe (Tektronix 6015A, 1000:1, Shanghai, China) and a voltage probe (PVP2150, RIGOL, Beijing, China) were used to record the applied high-voltage and voltage across the external capacitor, respectively. Both voltage signals were monitored using an oscilloscope (DS5062MA, RIGOL, Beijing, China). A capacitor (1 μF) was connected between the DBD reactor and the ground to measure the amount of transferred charge. The discharge power was controlled by varying the applied voltage across the plasma reactor, which was calculated using the standard Q-U Lissajous method (detailed information was provided in Text S3 and Figure S3 in the Supplementary Materials).

As for the gas analytical unit, NOx concentration was detected by a flue gas analyzer (Testo 340, Black Forest, Germany), with a resolution of 0.1 ppm and measurement accuracy of ±5%. Both inlet and outlet VOC concentrations were recorded online using a photo ion detector (PID, RAE 3000, Honeywell, Morris Plains, NJ, USA), with a resolution of 0.1 ppm and accuracy of ±3% in the TVOC value from 10 to 2000 ppm. VOC components analysis was performed by thermal desorption using a sorbent tube. Specifically, a commercial stainless-steel sorbent tube (TD100xr, Markes International, Bridgend, UK) packed with a carbon molecular sieve was used for the in situ collection of sinter flue gas VOCs. The gas was collected under a consistent flow rate of 50 mL/min for 30 min and then analyzed with a GC-MS system (7890B-5977B, Agilent, Santa Clara, CA, USA). Before each use, the sorbent tube was conditioned by 300 °C thermal cleaning with N$_2$ (purity ≥ 99.999%). In addition, a blank tube was analyzed before running the sample tubes. The absorption liquid anion (SO$_4^{2-}$, SO$_3^{2-}$, NO$_3^-$, NO$_2^-$, and Cl$^-$) was characterized using ion chromatography (IC 883, Metrohm, Herisau, Switzerland). The tail gas ozone concentration was monitored by an ozone monitor (Model 106, 2B, CO, USA) with detection accuracy of 1%.

*2.3. Plasma Status Determination and Calculation*

The discharge power employed for VOC and NO conversion was valued by applied peak voltage and specific input energy (SIE, J/L). The discharge power can be controlled by adjusting the applied voltage through the amplifier, with the input discharge power varying from 4.3 to 27 W, corresponding to SIE varying from 31.6 to 633.2 J/L (shown in Table S1 in the Supplementary Materials).

The efficiencies of the NO removal rate ($\eta_{NO}$), NOx removal ($\eta_{Nox}$), and TVOC removal efficiency ($\eta_{TVOC}$) were calculated using the following Equations (1) to (3):

$$\eta_{NO} = \left( \frac{[NO]_{in} - [NO]_{out}}{[NO]_{in}} \right) \times 100\% \tag{1}$$

$$\eta_{NOx} = \left( \frac{[NOx]_{in} - [NOx]_{out}}{[NOx]_{in}} \right) \times 100\% \tag{2}$$

$$\eta_{TVOC} = \left( \frac{[TVOC]_{in} - [TVOC]_{out}}{[TVOC]_{in}} \right) \times 100\% \tag{3}$$

## 3. Results and Discussion

### 3.1. In-Plasma Catalytic Oxidation of Nitric Oxide

NO is the major NOx component in sinter flue gas, which accounts for over 95% depending on the sinter raw mix and sinter bed permeability [47]. In addition, desirable NOx removal requires high NO-to-$NO_2$ conversion and effective sequential $NO_2$ absorption efficiency. After NO is quickly oxidized by oxidants produced from the plasma catalytic oxidation process, $NO_2$ exists as the main component of NOx. Therefore, NO oxidation efficiency plays a leading role in NO remediation owing to the desirable $Na_2SO_3$ scrubbing of $NO_2$.

NO conversion, $NO_2$ generation, and electron temperature under in-plasma catalysis were investigated. Pure $N_2$ and $O_2$ gas streams were well premixed with a volume ratio of 80:20 in a mixing chamber prior to the plasma catalytic region, giving a fixed inlet NO concentration of 200 ppm and flux of 400 mL/min without specific illustration. The electron temperature was monitored by an infrared thermometer (UT300S, Uni-Trend Technology CO., Ltd., Dongguan, China) by measuring infrared energy radiated from the high-voltage or discharge barrier surface. Under ideal discharge conditions, it was expected that NO would exhibit a higher conversion rate and result in relatively low $NO_2$ production. As shown in Figure 2, the NO conversion rate significantly improved to 92% when discharge power was above 6.75 W, and the concentration of $NO_2$ and electron temperature also increased, indicating the generation of energetic electrons by the electric field injection. When the discharge power was less than 10 W, the plasma discharge was not complete. After that, NO conversion was lower than 30%, and $NO_2$ generation reached about 120 ppm. After discharge power increased to 10 W, more active oxygen species were generated, thus leading to the NO removal efficiency increase by 93.3% and the corresponding $NO_2$ concentration of 125.3 ppm. When discharge power increased to 15 W, maximum NO conversion was 97.1%, and $NO_2$ concentration was 169.5 ppm. The electron temperature also increased to 50.4 °C. When discharge power increased to above 17 W, $NO_2$ generation and electron temperature increased obviously, and the NO conversion sharply decreased. Under complete discharge conditions, NO can be effectively oxidized to $NO_2$ by plasma-induced oxidants via a series of reactions [48,49]. The increasing amount of $NO_2$ was attributed to the plasma discharge of $N_2$ oxidation under an air atmosphere.

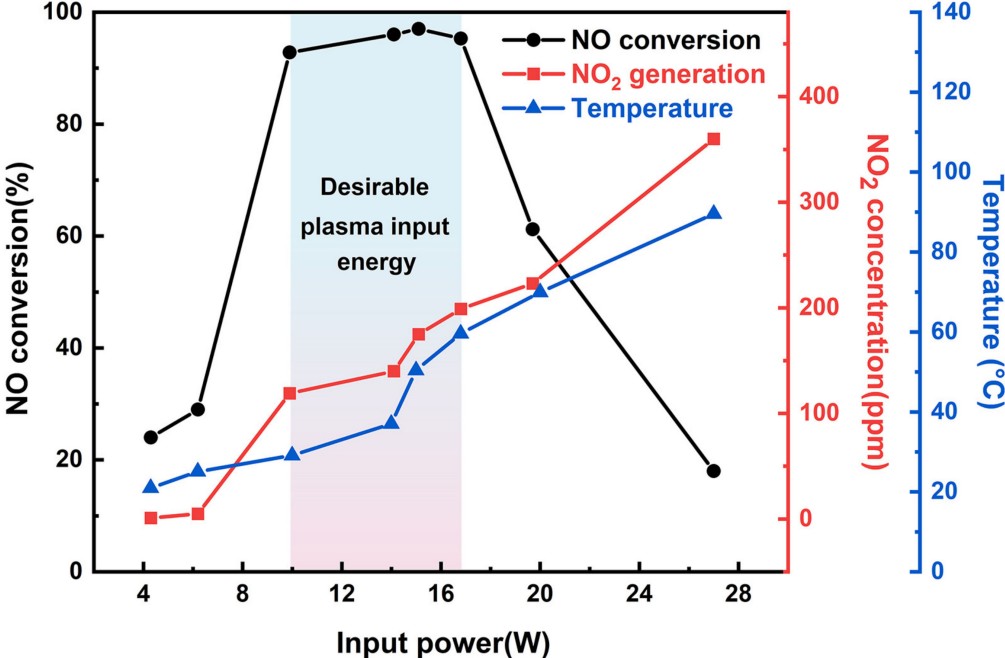

**Figure 2.** NO conversion (black), $NO_2$ generation (red), and electron temperature (blue) during a plasma catalytic oxidation reaction.

To sum up, depending on the NO conversion to $NO_2$ under different plasma power, the desirable discharge power range was between 10 to 17 W, the NO conversion rate reached $95.7 \pm 1.47\%$, and the corresponding $NO_2$ concentration reached $157.3 \pm 27.4$ ppm.

### 3.2. NOx Removal by IPC Coupling with $Na_2SO_3$ Scrubbing

3.2.1. Investigation on $Na_2SO_3$ Initial Concentration

The effective reaction of $SO_3^{2-}$ with $NO_2$ mainly depends on the $Na_2SO_3$ concentration in wet scrubbing reaction. A total of 0.5% of ascorbic acid was initially added to $Na_2SO_3$ absorption liquid to enhance the reducibility of the scrubbing process for each test. By fixed discharge power of 15 W, the NO conversion rate and corresponding $NO_2$ generation under different $Na_2SO_3$ concentration absorption were investigated. As shown in Figure 3a, the NO conversion rate remained at more than 95% after 120 min, when $Na_2SO_3$ concentration varied from 0.5% (wt.) to 5%. In addition, the increasing concentration of $Na_2SO_3$ exhibited a slight improvement in NO conversion. However, $Na_2SO_3$ concentration had a significant influence on $NO_2$ generation (Figure 3b). When the $Na_2SO_3$ concentration was 0.5%, after 45 min, the $Na_2SO_3$ scrubbing reaction was invalid. The increase in $Na_2SO_3$ concentration from 1% to 5% led to a longer capable absorption of $NO_2$ after 120 min. Increasing $SO_3^{2-}$ concentration exhibited a promotion effect on $NO_2$ absorption efficiency due to the reaction of $SO_3^{2-}$ and $HSO_3^-$ with $NO_2$ molecules [50].

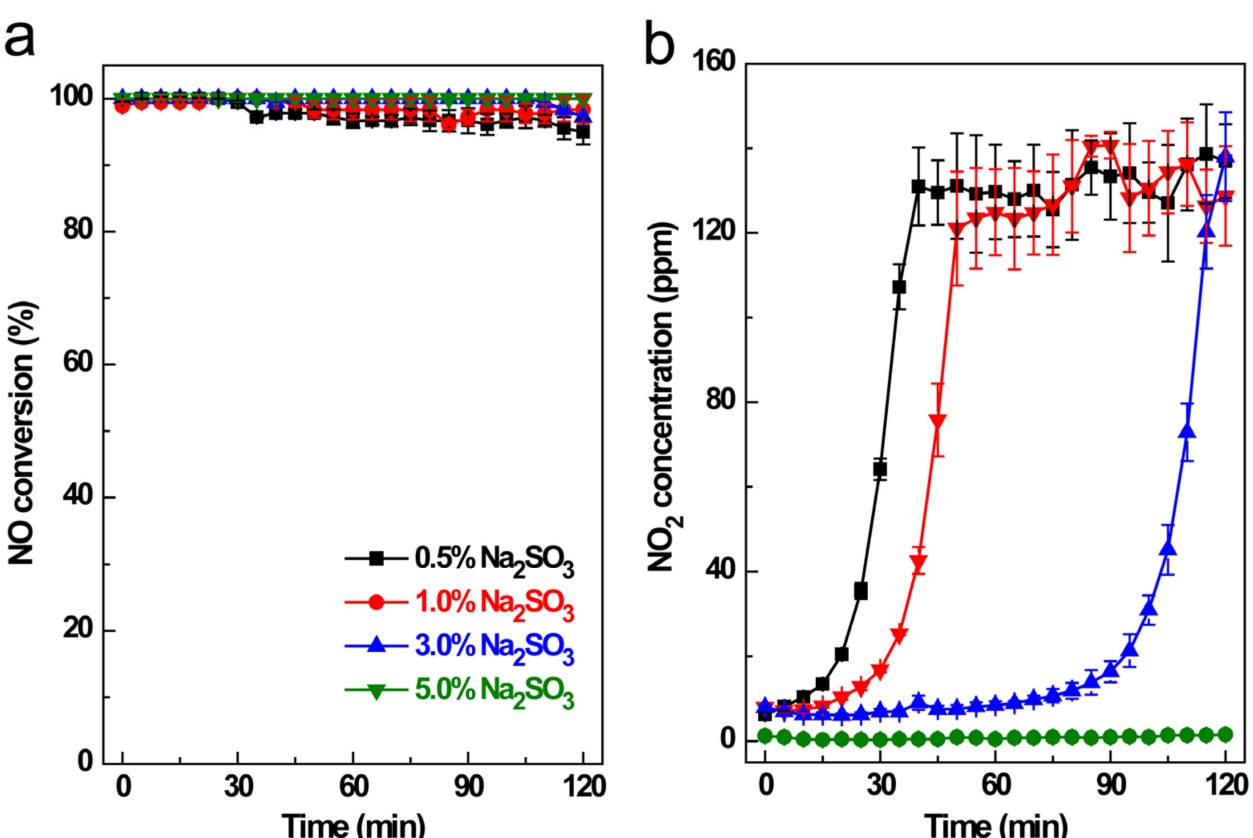

**Figure 3.** Effects of $Na_2SO_3$ initial concentration on NO conversion (**a**) and $NO_2$ generation (**b**) (input plasma power = 15 W, NO inlet concentration = 200 ppm, gas velocity = 500 mL/min, pH = 10).

To further confirm the $Na_2SO_3$ scrubbing mechanism, the absorption liquid was analyzed using ion chromatography, and the results are shown in Figure 4. It was obvious that $SO_4^{2-}$, $NO_3^-$, and $NO_2^-$ were major anions in the absorption liquid. However, the characteristic peak of $SO_3^{2-}$ does not appear in ion chromatography due to the ion chromatographic column (IonPac, AS22) not being able to separate $SO_3^{2-}$ and $SO_4^{2-}$. With the $Na_2SO_3$ initial concentration increasing from 0.5% to 5%, the $SO_4^{2-}$ concentration in the absorption liquid

dramatically increased. It can be observed in Figure 3 that $SO_3^{2-}$ was completely consumed by the reaction with $NO_2$ and $O_3$ when the $Na_2SO_3$ initial concentration was less than 3%. When 5% of $Na_2SO_3$ was introduced, none of the $NO_2$ concentration was detected in the gas effluent, indicating that the excess of $SO_3^{2-}$ remained in the scrubbing solution. With the increasing $Na_2SO_3$ concentration, $NO_2^-$ concentration increased. According to Equation (4), the enhancement of $SO_3^{2-}$ in the absorption liquid was favorable for $NO_2$ conversion to $NO_3^-$ and $NO_2^-$ due to the improving alkalinity and reducibility. However, $NO_3^-$ concentration was relatively constant after different $Na_2SO_3$ absorption. A total of 1% of $Na_2SO_3$ was chosen for the following experiment.

$$2NO_2 + H_2O + 2SO_3^{2-} \rightarrow NO_2^- + NO_3^- + 2HSO_3^- \tag{4}$$

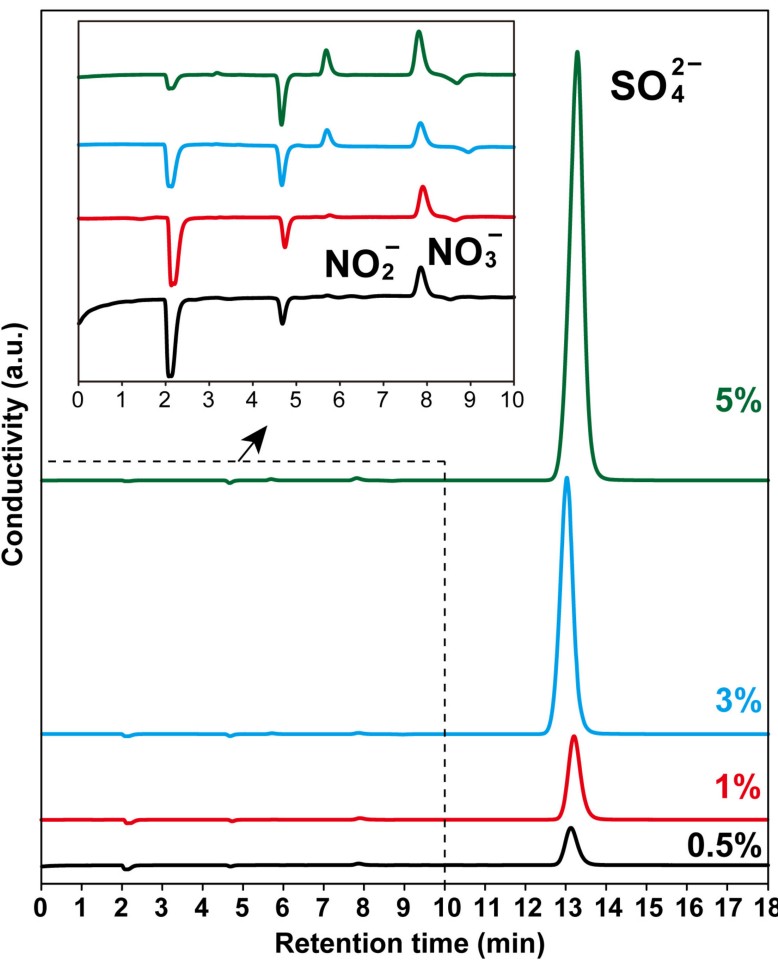

**Figure 4.** Ion chromatography of $Na_2SO_3$ absorption solution after 120 min reaction under different $Na_2SO_3$ concentrations.

### 3.2.2. Investigation of pH Value

The influence of $Na_2SO_3$ solution pH value on both NO conversion and $NO_2$ generation is shown in Figure 5a. Compared to the effects of $Na_2SO_3$ concentration, pH value had a slight influence on NO conversion but exhibited different affection on $NO_2$ generation. The enhancing acidity of the $Na_2SO_3$ solution inhibited NO conversion to some extent. Specifically, NO conversion decreased to 91.3% after 40 min absorption under a pH value of 5. For a pH value of 7, the NO conversion was 100% when absorption started and decreased to 91.5% after 30 min. When the pH value increased to 10 and 12, NO conversion was 100% at 35 min after the reaction and remained at 94.3% and 97.5% after 120 min, respectively.

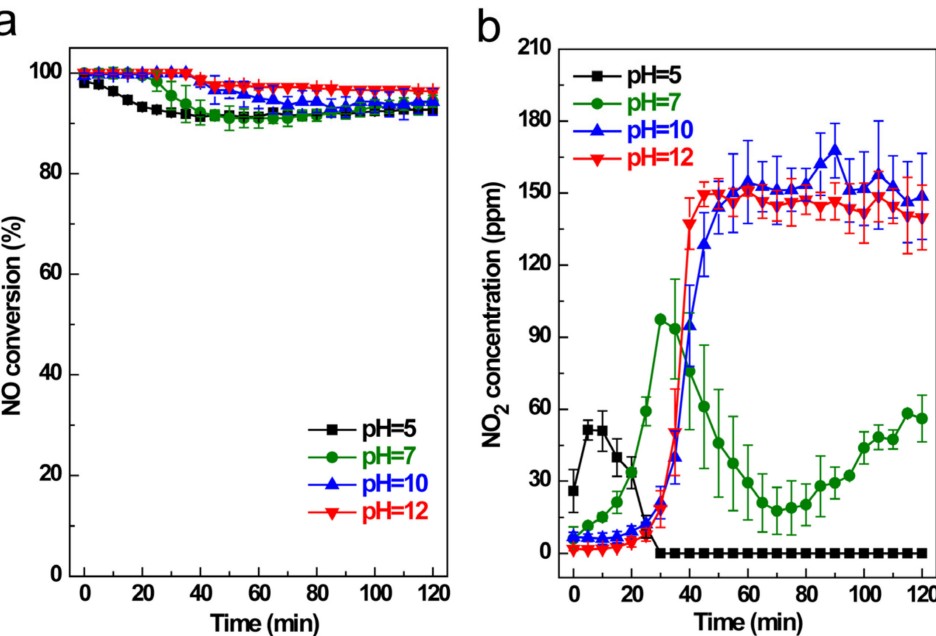

**Figure 5.** Effects of $Na_2SO_3$ solution pH value on NO conversion (**a**) and $NO_2$ generation (**b**). (Input plasma power = 15 W, NO inlet concentration = 200 ppm, gas velocity = 500 mL/min, $Na_2SO_3$ concentration: 1% wt.).

The influence of $Na_2SO_3$ solution pH value on NO conversion can be further analyzed by outlet $NO_2$ concentration. Figure 5b shows that outlet $NO_2$ concentration fluctuated obviously under different pH values. When the pH value was 5, $NO_2$ increased to 51.4 ppm and decreased from 10 min. No $NO_2$ was detected after 35 min of reaction. When the pH value was 7, $NO_2$ generation increased to 100 ppm after 30 min and then decreased from 30 min to 70 min. In the acid $Na_2SO_3$ solution, the weak reducibility resulted in a fast $SO_3^{2-}$ consumption by $NO_2$. After that, the outlet $NO_2$ concentration in the first 30 min increased under pH = 5 and pH = 7. In addition, $SO_2$ was generated after 30 min of reaction, suggesting that by promoting the acidity of $Na_2SO_3$ absorption, the existing abundant $H^+$ reacted with $SO_3^{2-}$ facilitated the reaction of $SO_2 + NO_2 \rightarrow SO_3 + NO$, thus leading to a decrease in $NO_2$ concentration and NO conversion. When the pH value increased to 10 and 12, obvious $NO_2$ emission was observed after 30 min absorption, suggesting $SO_3^{2-}$ had been completely consumed, and the outlet $NO_2$ concentration remained at 140~150 ppm throughout 120 min of reaction.

The absorption liquid under different pH values was also investigated by ion chromatography, as shown in Figure 6. By fixing $Na_2SO_3$ concentration of 1%, it was apparent that $SO_4^{2-}$ peak intensity was approximate under different pH values. The obvious difference was $NO_3^-$ and $NO_2^-$, where the pH value increased from 5 to 12. When the pH value was less than 7, only $NO_3^-$ and $SO_4^{2-}$ existed in the scrubbing solution, which indicated the invalidity of $SO_3^{2-}$ after 120 min reaction. With the pH value increased to 10 and 12, obvious $NO_2^-$ was detected, indicating that the reducibility of the scrubbing solution can promote the reaction of $SO_3^{2-}$ with $NO_2$. The following experiment fixed the $Na_2SO_3$ solution pH value of 10.

### 3.2.3. Investigation of NO Velocity

Depending on different sinter conditions, the NO velocity in sinter flue gas may fluctuate due to the gas permeability by air suction in the sinter bed [47]. The influence of gas velocity was also an important parameter that influenced removal efficiency. As shown in Figure 7a, IPC coupling with $Na_2SO_3$ scrubbing exhibited desirable NOx removal under different NO velocities. When NO velocity increased from 400 mL/min to 1500 mL/min, the $NO_2$ concentration after IPC coupling $Na_2SO_3$ scrubbing showed a slight increase

(Figure 7b), and the NOx removal rate showed a declining trend. The NOx removal rate was more than 70% under a velocity of 1500 mL/min, and the corresponding $NO_2$ concentration was less than 15 ppm. The thorough treatment efficiency of NO under different velocities can be attributed to the effective oxidation capacity of the in-plasma catalytic oxidation of NO molecules.

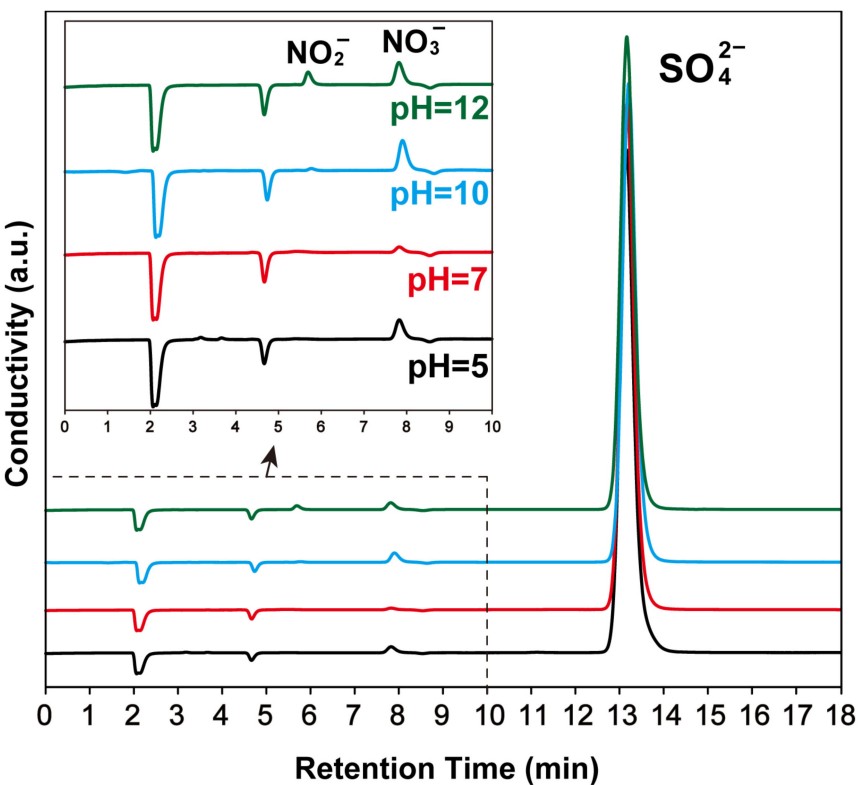

**Figure 6.** Ion chromatography (IC) of $Na_2SO_3$ absorption solution after 120 min reaction under different pH values.

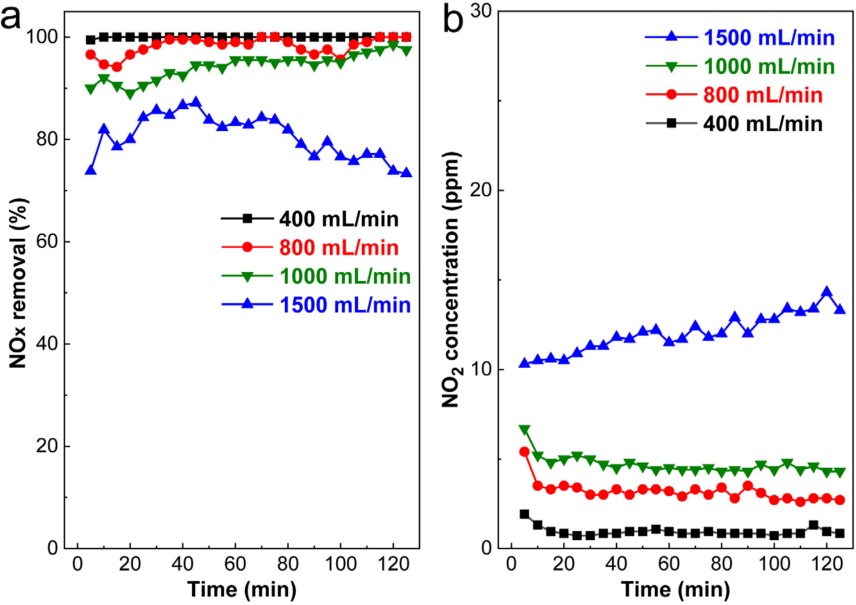

**Figure 7.** Effects of NO velocity on NOx removal (**a**) and outlet $NO_2$ concentration (**b**) (NO initial concentration = 200 ppm, discharge power = 15 W, $Na_2SO_3$ = 1%, pH value = 10).

### 3.2.4. Investigation of NO Initial Concentration

The NO concentration in industrial sinter flue gas is often variable due to the uneven distribution of fossil-fuel nitrogen and incomplete combustion [51]. The different sinter raw mix and ratio also affect NO emission [52]. Herein, the influence of NO initial concentration on NOx removal was investigated, and the results are shown in Figure 8. The outlet $NO_2$ concentration after IPC coupled $Na_2SO_3$ absorption kept less than 35 ppm when NO initial concentration was 100 to 600 ppm. The inlet NO concentration had an obvious influence on NOx removal efficiency. NOx removal decreased with the increase in NO inlet concentration. When NO inlet concentration was 100 ppm, 300 ppm, 450 ppm, and 600 ppm, the corresponding NOx removal efficiencies after 120 min reaction were 100%, 44.5%, 23%, and 4%, respectively. The inlet NO concentration determined the NO molecules amount throughout the plasma region per unit of time. Under the same discharge power, the reactive oxygen species ($O_3$, ·OH, ·O, etc.) were consistent. After that, more NO molecules may not be effectively oxidized to $NO_2$ with the increase in NO concentration, which resulted in an obvious decrease in NOx removal efficiency.

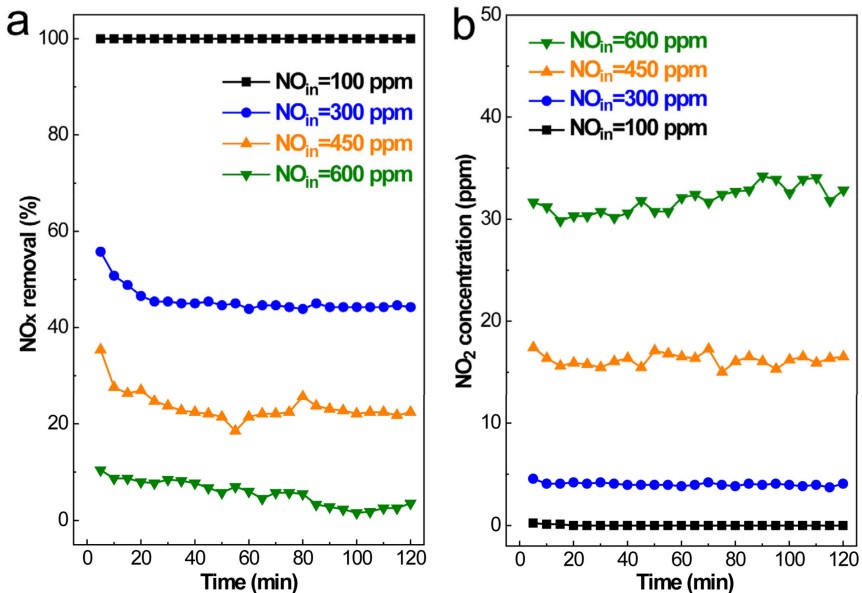

**Figure 8.** Influence of NO initial concentration on NOx removal (**a**) and outlet $NO_2$ concentration (**b**) (NO velocity = 200 mL/min, discharge power = 15 W, $Na_2SO_3$ = 1%, pH value = 10).

It can be observed from Figure 8b that the outlet $NO_2$ concentration after treatment of IPC coupling with $Na_2SO_3$ absorption can be effectively reduced, which is less than 35 ppm when inlet NO concentration varies from 100 to 600 ppm. In conclusion, NO concentration showed a greater influence than that of NO velocity.

### 3.3. Co-Elimination of NO and VOCs after IPC Combined with $Na_2SO_3$ Wet Scrubbing
### 3.3.1. Removal of NO

First, the sinter raw mix was heated to 400, 450, and 500 °C in the tubular furnace using air as a carrier gas, thus providing simulated sinter flue gas with different NO inlet concentrations. It should be noted that under fixed calcination temperature, the simulating sinter flue gas NO concentration was not stable due to fuel-N combustion. As shown in Figure 9a, under calcination temperatures of 400, 450, and 500 °C, the corresponding maximum NO concentrations were 120 ppm, 304 ppm, and 450 ppm. When the sinter temperature was 400 °C, the NO produced by the sinter raw mix followed an increasing trend for 20 min and then kept steady until 120 min. Meanwhile, no NO was observed in the outlet after IPC and $Na_2SO_3$ scrubbing, indicating that NO inlet within 120 ppm can be effectively eliminated. When the sintering temperature increased to 450 and 500 °C, the NO

concentration generated by fuel-N combustion fluctuated, showing an increasing trend, followed by a decrease after 30 min. We further calculated the NO removal efficiency corresponding to NO inlet concentration varying from 100 ppm to 450 ppm (Figure 9b). When the NO inlet concentration was less than 200 ppm, the NO removal rate was more than 80%. In addition, the NO removal rate decreased to 60% when the NO inlet concentration was 450 ppm. The above results indicated that high NO concentration has a suppressive influence on the NO removal rate, while IPC coupled with $Na_2SO_3$ scrubbing for practical NO treatment showed a wide application range of different NO concentrations.

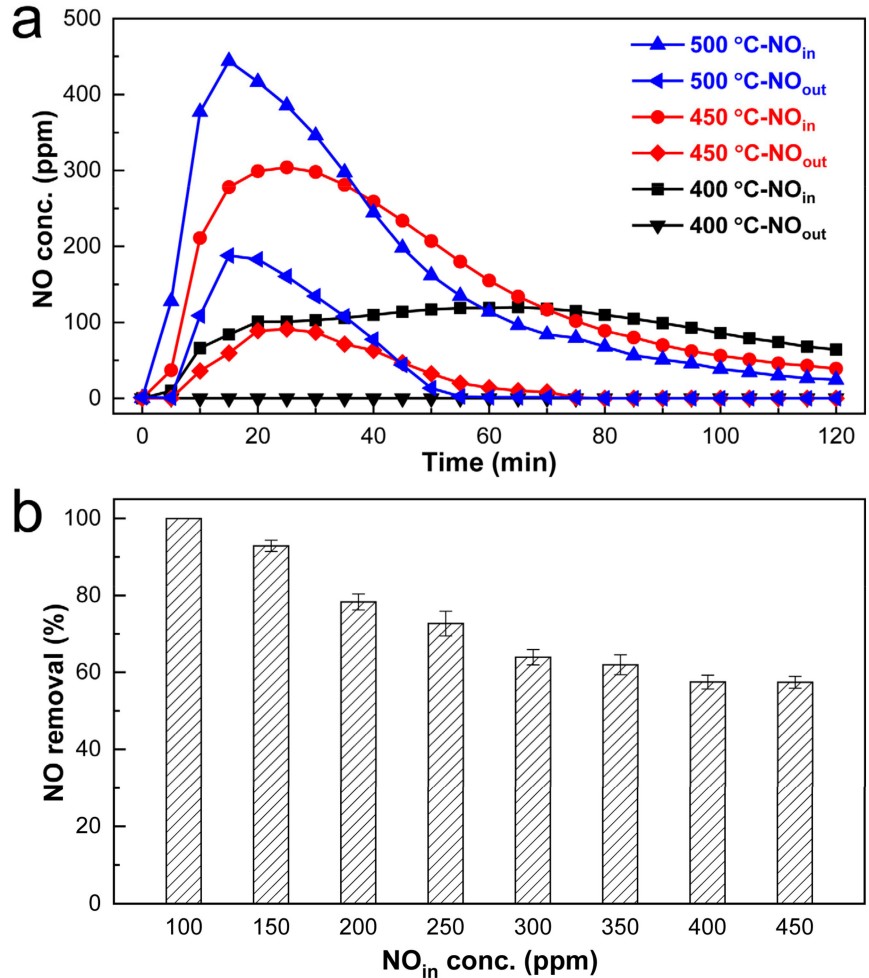

**Figure 9.** NO removal by IPC coupled $Na_2SO_3$ absorption (under different sinter time and sinter temperature (**a**); under different NO inlet concentrations (**b**)).

### 3.3.2. Removal of VOCs

VOCs can be produced simultaneously by the heating of sinter raw mix in a fixed airflow. TVOC emission showed a similar trend, with NO concentration under different sinter temperatures, as shown in Figure 10a. With the increasing sinter temperature, the TVOC emission showed an obvious enhancement from 400 °C to 450 °C. The maximum TVOC was 14.9 ppm, 58.6 ppm, and 82.2 ppm, corresponding to sinter temperatures of 400, 450, and 500 °C. The TVOC removal efficiency was calculated by $TVOC_{in}$ and $TVOC_{out}$ using a PID detector. IPC coupling with $Na_2SO_3$ wet scrubbing showed remarkable performance for sinter flue gas VOC elimination under different sinter temperatures (Figure S4 in the Supplementary Materials). The TVOC removal efficiency appeared to have a slight descending trend when TVOC initial concentration varied from 20 to 80 ppm (Figure 10b), with 99% at 20 ppm and 85.7% at 80 ppm. $Na_2SO_3$ wet scrubbing can effectively absorb the excess $O_3$ and dissolved organic molecules produced by the plasma catalytic reaction.

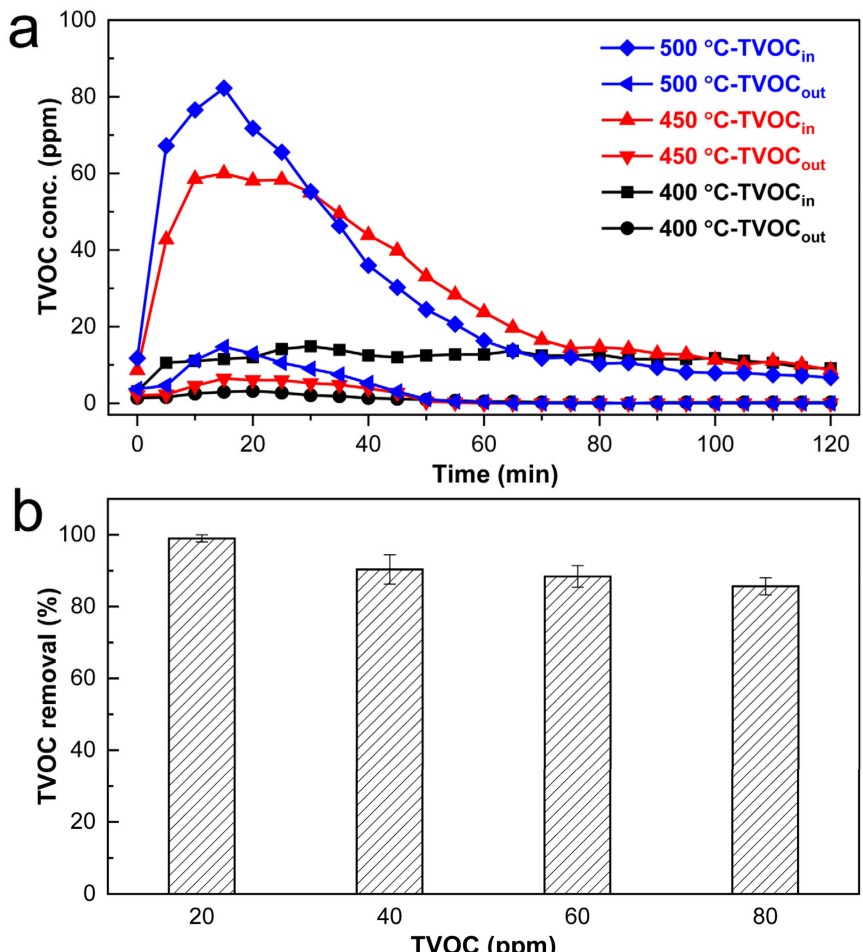

**Figure 10.** TVOC removal by IPC coupling with $Na_2SO_3$ scrubbing: under different sinter times and sinter temperatures (**a**); under different VOCs inlet concentration (**b**).

To investigate the VOC removal efficiency by the coupling system, we further conducted TG-GC-MS analysis. The simulated sinter flue gas VOC components were collected and concentrated using a TD100xr sorbent tube under 450 °C and a flow rate of 50 mL/min for 30 min and then analyzed by TG-GC-MS analysis. As shown in Figure 11, the detected VOC components before treatment comprised more than 50 species, of which BTEX was the dominant component. The results were in accordance with our previous research [22]. Distinctive VOC reduction was observed after the IPC with $Na_2SO_3$ scrubbing treatment, suggesting the satisfactory removal efficiency of VOCs, which can also be proved by Figure 10b with TVOC removal > 88% under 450 °C. The detected VOCs were only methyl-cyclohexane, toluene, p-Xylene, propylbenzene, 1,4-dichloro-benzene, and benzaldehyde, with an obvious decreasing chromatographic peak intensity, respectively. However, the increasing peak intensity of benzene indicated that benzene was the major degradation byproduct. The VOC removal results indicated that IPC with $Na_2SO_3$ scrubbing removes the features of flue gas with NO and VOC components and large gas flux.

### 3.4. Role of $Na_2SO_3$ Scrubbing

To investigate the NOx conversion route and elimination mechanism via IPC combined with the sequential $Na_2SO_3$ scrubbing treatment, we analyzed different absorption solutions after 30 min of IPC reaction using ion chromatography (IC-883, Metrohm). Before analysis, simulated sinter flue gas was obtained by mixing 300 ppm $SO_2$ with sinter flue gas produced by 450 °C heating of sinter raw mix in a tubular furnace. As shown in Figure 12, the major anions by pure water absorption for IPC pre-oxidation of simulating sinter flue gas were

$SO_3^{2-}$, $SO_4^{2-}$, and $F^{-1}$. They can be attributed to $SO_2$ and fluoride dissolution and further oxidation by reactive oxidation species generated by plasma catalysis. The $NO_3^-$ peak after water absorption can be clearly observed when the simulated sinter flue gas was treated via an in-plasma catalytic region, indicating the efficient oxidation of NO to $NO_2$. When using 1% $Na_2SO_3$ as a scrubber, only $SO_3^{2-}$ and $SO_4^{2-}$ were observed in the absorption liquid. In addition, the peak intensity of $SO_4^{2-}$ was obviously higher than that of $SO_3^{2-}$ due to the excess ozone oxidation, which was generated by plasma. It should be noted that the absorption liquid of 1% $Na_2SO_3$ was diluted 100-fold compared to that of water as a scrubber.

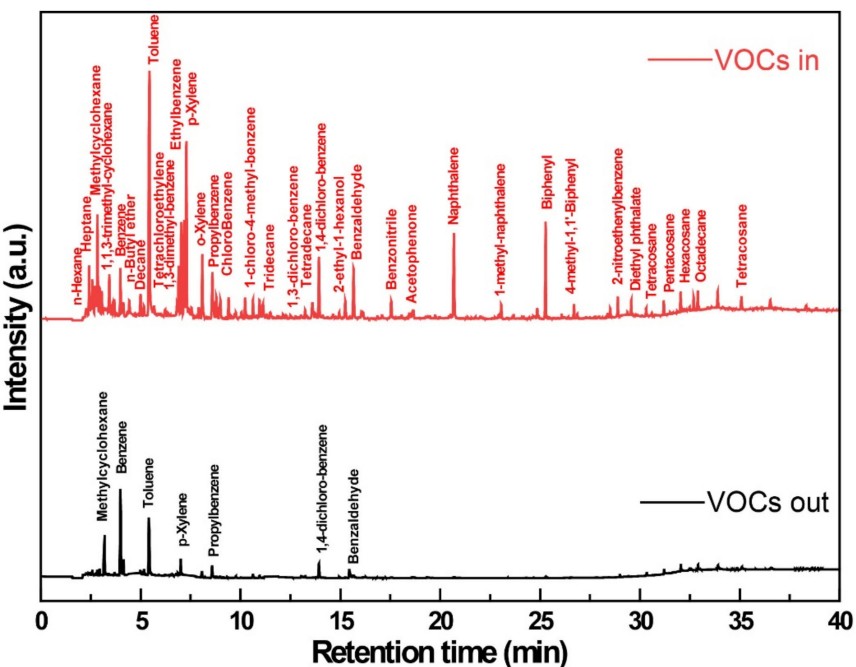

**Figure 11.** TG-GC-MS characterization of sinter flue gas VOC components before and after IPC coupled $Na_2SO_3$ scrubbing.

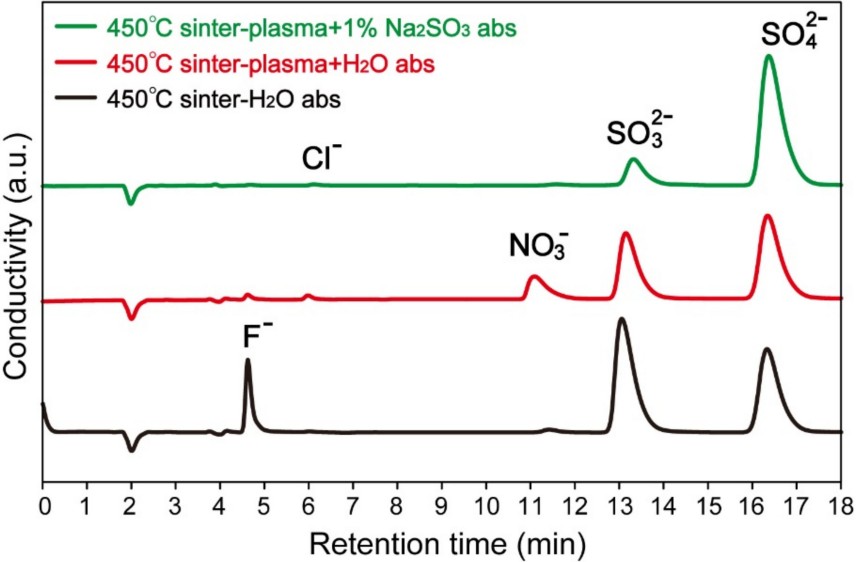

**Figure 12.** Ion chromatography of sinter flue gas different absorption solutions before and after IPC treatment.

## 4. Conclusions

The enormous and complex air-pollutant emissions from the iron and steel industry place a huge burden on China's regional atmospheric environment. The integration of in-plasma catalysis with sequential $Na_2SO_3$ treatment can be effective for the co-elimination of sinter flue gas multi-components. The plasma discharge status was optimized by investigating NO conversion. The VOC and NO removal performance of the integrated system was further investigated by taking simulated sinter flue gas as model pollutants. NO removal rate was more than 80% when the initial concentration was less than 200 ppm. In addition, 88% of the TVOC removal rate can be realized when the TVOC concentration is no more than 80 ppm. The findings indicate that plasma catalysis integrates with $Na_2SO_3$ scrubbing for a collaborative effect in the co-elimination of sinter flue gas multi-compound emissions. The future investigation of the proposed technology should consider the nitrogen compound balance and the evaluation of practical sinter flue gas. The development of a plasma catalyst and the optimization of DBD reactor geometry can also promote the application of the combining system.

**Supplementary Materials:** The following supporting information can be downloaded at https://www.mdpi.com/article/10.3390/pr11102916/s1, Figure S1: DBD reactor with wedged high-voltage electrode: figure illustration (a) and photograph (b); Figure S2: Schematic diagram of experimental setup for plasma status detection; Figure S3: Output voltage signals of plasma discharge (a) and Lissajous figure (b) at 15 kV, 7.5 kHz; Figure S4: Comparison of different treatment processes on sinter flue gas TVOC removal; Table S1: Specific input energy (SIE) with different input power and peak voltage. Text S1: Description of DBD reactor; Text S2: Detailed plasma status detection; Text S3: Discharge power calculation. Text S4: Investigation on different treatment process on sinter flue gas TVOC removal.

**Author Contributions:** Conceptualization, methodology, investigation, writing—original draft, funding acquisition, J.L.; investigation, data curation, R.Z. and M.S.; visualization, validation, Q.S. and M.Z.; project administration, funding acquisition, J.Z.; writing—review and editing, Y.L.; supervision, J.J. All authors have read and agreed to the published version of the manuscript.

**Funding:** This research was supported by the National Natural Science Foundation of China (No. 22306211) and the Natural Science Foundation of Henan (No. 212300410322). Financial support from Zhongyuan University of Technology Natural Science Foundation (No. K2022QN027), Zhongyuan University of Technology Postgraduate Education Quality Improving Project (No. JG202217, ALK202309), Zhongyuan University of Technology Discipline Strength Improving Project (No. SD202242) and College Students Innovation Training Program of China (202310465033) were also acknowledged.

**Data Availability Statement:** Data will be made available upon request from the corresponding author. The data are not publicly available due to privacy.

**Acknowledgments:** We appreciate the sinter raw material supply from Taiyuan Iron & Steel Co., Ltd.

**Conflicts of Interest:** The authors declare no conflict of interest.

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
