# Peer review of "Collaborative Effect of In-Plasma Catalysis with Sequential Na2SO3 Wet Scrubbing on Co-Elimination of NOx and VOCs from Simulated Sinter Flue Gas"

_processes, doi:10.3390/pr11102916_

Round 1

Reviewer 1 Report

The information data are interesting but the presentation must be improved.

All the figures must be plotted from the software source. It seems that all the figures are pictures of low resolution, such as Copy/Paste from another document.

Error bars are required too for experimental data.

Additional comments:

1. What is the main question addressed by the research?

Coupling the plasma-catalyst reactor with Na2SO3 as a scrubber could be an interesting idea for VOC removal from gas.

2. Do you consider the topic original or relevant in the field?

Does it address a specific gap in the field? The topic has already been addressed in the literature. Originality may come from the treatment of a specific gas resulting from a specific technological process.

3. What does it add to the subject area compared with other published material?

In order to answer objectively on the novelties brought by this treatment system, the contribution of the plasma catalyst reactor in the balance of the treated chemical compounds should first be highlighted. A comparison of the scrubber with the NTP-catalyst reactor + scrubber could highlight the synergistic effect of the two methods considered.

4. What specific improvements should the authors consider regarding the methodology?

What further controls should be considered?

Which plasma reactor parameters have the main effect of removing VOCs

5. Are the conclusions consistent with the evidence and arguments presented and do they address the main question posed?

Yes

6. Are the references appropriate?

Can be improved.

7. Please include any additional comments on the tables and figures. The figures have a very poor resolution and some are illegible. They need to be redone before publication.

Author Response

Dear reviewer, we are appreciated for the suggestive comments, which help us to enhance the articel quality and give direction for future study. After carefully checking all the comments on our manuscript, we are resubmitting the revised manuscript and enclosed response file.

Reviewer 2 Report

This experimental work on an air purification system includes plasma catalysis combined with a Na2NO3 solution. There are several major issues in the manuscript that need revision. Details are below.

First, the authors are focusing on the roles of DBD power and the ion concentrations in the Na2NO3 stage. Apparently, the final NOx removal rate depends on the reaction rates in the pathways of these two stages. However, we see no analysis and discussions with quantifications of these reaction rates and rate coefficients. There are merely limited discussions in text only. The rate coefficients are functions of reactant temperatures. There are both exothermic and endothermic reactions between the NOx species and the solution, as well as the dissolution process of gases. Also, downstream of the DBD reactor, the gases are heated up, but they are also cooling down during the travailing in the tubing. The temperature monitor is also missing which needs to be added. The temperature depends on the exact geometry of the tubing, flow rate, initial temperate in the DBD, and also the composition. Of course, the initial temperature in the DBD depends on the plasma status, indirectly controlled by the power.

This leads to the second issue in the manuscript. The description of plasma status is not effective at all. DBD discharge contains numerous filaments and each of them is a streamer propagation. This is a thermal non-equilibrium plasma with an electron temperature very different from other species’ temperatures. The plasma chemistry in DBD is initially triggered by the electron impacts, the rate coefficients of the initial reactions in the plasma chemical pathways are functions of the electron temperature which is a function of the local electric field. The electric field is a spatial gradient of the electric potential and it can only be solved through the Poisson equation. Therefore, the electron temperature is determined not only by the gas composition input, but also by the geometry of electrodes, discharge frequency, and the discharge voltage. The discharge power is thus not sufficient to describe such a complicated plasma status. Using discharge power to represent the working condition of the DBD reactor is meaningless. The authors need to reconsider and add more monitoring of plasma parameters in the plasma stage.

This leads to the third problem. How is the temperature controlled after the DBD stage? The authors have temperature quantified and shown in figures. But those are the sinter temperatures. The geometry of the tubing system surely determines the heat dissipation of the gas flow and the resulting gas temperatures in the reactors determines the reaction rate coefficients. There are no measurements and controls on these parameters. This problem is fatal.

Other minor issues include the low resolution of all the figures, especially Figure 1; and the improper terminology of NOx removal rate. The removal rate is defined as dnNOx/dt which is the left-hand-side of a rate equation.

Overall, this work is more like a results report which shows us how the hardware setups can determine the final results. There are no quantifications and detailed discussions on the physical and chemical mechanisms behind such an input-output relation of the entire reactor system. It is surprising that research work focusing on chemical processing has no discussions on the reaction rates. The authors also completely missed the monitoring and controls of the gas flow temperature and the temperatures in the reactor stages, as well as the plasma parameters. Therefore, they do not have any information about the reaction rates which directly determine the final results. Considering these significant issues, and the additional controls and measurements required, I cannot give more positive comments but have to suggest a rejection of this work. 

The quality of English is mostly fine. 

Author Response

(The authors gave the same response as above.)

Reviewer 3 Report

Dear authors,

The article is of high quality and in my opinion the subject of research is suitable for publication in a journal. However, some issues need to be corrected to make the article more understandable for readers of the magazine. In my opinion, the title of the article should be changed to indicate that the article presents the results of research on a laboratory scale. leaving the title in its current form suggests that we are dealing with industrial research. Below is a list of issues to be clarified and specified in the article:

1. During the research, did you notice the reduction of NO2 to N2 in the Na2SO3 solution?

2. Have you performed an ion balance for the process, if so, it is worth including these results in the article, with information whether the balance of nitrogen compounds is closed.

3. In the "materials and methods" section, the uncertainties of the measuring instruments used should be described.

4. In the summary section, the plan for future research should be described.

Best regards for authors.

Author Response

(The authors gave the same response as above.)

Round 2

Reviewer 1 Report

Thank you for your answers

Author Response

Dear reviewer, thanks for your affirmation on our revised manuscript. We will promote the study and make indepth findings.

With best regards.

Reviewer 2 Report

The author didn’t make a point-to-point response in the response letter. My previous comments include several paragraphs and each of them focused on a major point. The authors should follow the general format of a response letter and respond to each sentence or, at least, each paragraph in the comments. At the current stage, the response letter is not effective.

The main and fatal issue is not responded and the authors have no correction in the manuscript about it. The role of plasma in the entire system is unclear. The whole reactor relies on the low-temperature plasma chemistry in the DBD stage. But the plasma status is not described at all. As I introduced in my previous comments, each of the rate coefficients in the chemical pathway depends on the species' temperatures, and the plasma chemical pathway is triggered by the electron-impact reactions where the coefficients depend on the electron temperature. The electron temperature is non-linearly proportional to the reduced electric field which depends on both the electric field and the particle density. Due to the existence of charges, the electric field does not simply equal the discharge voltage over the gap distance of DBD but has to be solved through the Poisson equation where the discharge voltage is the boundary condition. Above is how the discharge voltage is related to the chemical reaction rates in the system, along with the DBD hardware geometry, gas temperature, and pressure playing roles in the system. Therefore, simply using discharge voltage or power is not enough to describe the plasma status. But the authors have neither discussions about such a mechanism nor proper experimental design to show how the control parameters such as discharge voltage can affect the chemical rate coefficients in the reaction pathways. The authors even do not have any pathways shown in the manuscript. I am wondering if they really understand how the plasma works in the system. What are the short-lifetime species in the DBD stage and how do they react with the input gas? What are their reaction rate coefficients versus the temperature and time? What are the long-lifetime species generated in the DBD stage, and how do they react with the gas along the flow in the tubing? How are these short and long-lifetime species generated, and how their concentrations are related to the discharge voltage? These are some basic topics that should be discussed but the authors failed to discuss these topics.

This work is using a specifically designed plasma generator. Therefore, unless other researchers are using the exactly same generator, simply using the same discharge voltage and power setup is not possible to repeat their experiment and verify their results. In other words, the results in this work are not repeatable, unless the authors can provide detailed plasma parameters such as the electron density, electron temperature, and other species concentrations, etc. If they can provide these plasma parameters, other researchers can repeat this experiment with other DBD generators.

Overall, the manuscript is not showing a clear mechanism of the system, maybe the authors do not know the details of it. According to their response letter, they have difficulty investigating and are also not willing to. This leads to a rejection comment. The authors emphasized that the idea of this work is new but this is also why they must provide an effective description of the plasma status for other researchers to repeat and verify the results.

English quality looks fine. 

Author Response

Dear reviewer, thanks for your valuable comments on our Round 1 revised manuscript. We are sorry for the unclear response and misunderstanding of some topics in Round 1 commentary. Based on the commentary, we uploaded the revised version and response after careful revision. Please check the attached files.

With Best regards.

Reviewer 3 Report

Dear authors,

Thank You for detailed answers to my questions. Manuscript is suitable for publication in journal.

Best regards. 

Author Response

(The authors gave the same response as above.)

Round 3

Reviewer 2 Report

I would like to thank the authors for their response to my previous comments. But there are still problems.

The authors emphasized that their main novelty is the improved efficiency of VOCs and NOx removal. But this is exactly why the unclear description of plasma status is an issue. Is the DBD device in this work monopolized the market of cylinder-DBD? Are there any other similar devices that exist? Apparently, none of the other cylinder-DBD users, including those references cited in the response letter, use the exact same DBD generator. The distance between the electrodes determines the initial electric field of the discharge area. Therefore, as I emphasized in my previous comments, a different hardware geometry will lead to a different electron temperature and then, different chemical rate coefficients of all the electron-impact reactions and ion reactions. It is true that playing with the discharge voltage and flow rate can lead to different species removal results. But different cylinder-DBD sizes have different optimal voltages. Any changes to the outer diameter, inner diameter, length, etc. can alter the optimal control inputs. This is why I pointed out that this work is not informative because the authors improved their own device only, they don’t provide any general information for other cylinder-DBD users to adjust and optimize other cylinder-DBD generators. If the authors can provide the plasma parameters, such as the electron temperature, one can derive the optimal chemical rate coefficients and acquire the same improved results even with a different cylinder-DBD system. Please note that the type of power supply has nothing to do with the general idea of plasma chemistry optimization. The authors claimed that many other research teams also play with the discharge voltage and try to improve the results. Why the authors don’t simply use one of the voltage setups that were improved from other research works? Because, as I mentioned above, the other’s optimal voltage is not the optimal one here, due to the different discharge area geometry and even the different electrodes. While there have been too many similar research works on the market, we expect someone can dig deeper into the plasma behaviors in the reactor and provide a general optimization mechanism, not another paper of the improvement for the researcher’s own device.

It is good to see a list of reactions in the response letter and the supplementary document. But there are fundamental mistakes. The charge and elements are not balanced in those chemical reactions. For example, O2 + e => O + O is wrong. The electron disappeared on the right-hand side. The same issues can be found in many other reactions. Please also make sure that all the major reactions are included in the list. The plasma in the air can have hundreds of reactions. More importantly, the authors do not provide the reaction rate coefficients of these reactions. The rate coefficients are the most important key to dig deeper into this work. Because the rate coefficients are the functions of the reactant temperatures which are monitored by the authors. The gas bulk temperature is determined by the initial gas temperature, the momentum transfer from the electron temperature, and the heat dissipation. I expect the authors to investigate and discuss the rate coefficients and these are the variables actually connecting the final removal rate, the temperatures, and the voltage inputs. The discussion could also include several equations to show the math relations among them. This is the true value of such type of work which can provide the general mechanism of not only how the plasma works in the system, but also how to optimize it. The authors have added power computation in the response letter and the supplementary material, but similarly, they didn’t show the relation between the power and the results. The equations are the bridges to connect them. My suggestion is to show: voltage(power) with flow rate and initial gas status => electron and gas bulk temperatures => chemical reaction rate coefficients => reaction rates => chemical product composition.

Another issue is in Fig. S1. The authors use a voltage probe (the lower one) to measure the ground, which is ridiculous. The resulting voltage will surely be a constant zero volt. But surprisingly, they have a sinusoidal voltage signal shown in Fig. S2. Please double-check the experimental setup.

Overall, I would like to suggest either a resubmission after rejection or a major revision.

Author Response

Dear reviewer,

Thanks for your further comment on the revised manuscript and our Round 2 response. We have been working hard on the revisions according to your valuable suggestions and commentary. Please check the corresponding point-to-point response in the attached file. We sincerely hope you can consider the novelty of our research which targeted at collaborative elimination of sinter flue gas VOCs and NOx under ambient temperature.

With best regards.
